# Prenatal cadmium exposure does not induce greater incidence or earlier onset of autoimmunity in the offspring

Jamie L. McCall[1], Harry C. Blair[2,3], Kathryn E. Blethen[1], Casey Hall[1], Meenal Elliott[1], John B. Barnett[1,4]*

1 Department of Microbiology, Immunology & Cell Biology, West Virginia University School of Medicine, Morgantown, WV, United States of America, 2 Department of Pathology, Pittsburgh VA Medical Center, Pittsburgh, PA, United States of America, 3 Department of Cell Biology, the and the University of Pittsburgh, Pittsburgh, PA, United States of America, 4 West Virginia University Cancer Institute, West Virginia University, Morgantown, WV, United States of America

* jbarnett@hsc.wvu.edu

**Data Availability Statement:** All relevant data are within the manuscript and its Supporting Information files.

## Abstract

We previously demonstrated that exposure of adult mice to environmental levels of cadmium (Cd) alters immune cell development and function with increases in anti-streptococcal antibody levels, as well as decreases in splenic natural regulatory T cells (nTreg) in the adult female offspring. Based on these data, we hypothesized that prenatal Cd exposure could predispose an individual to developing autoimmunity as adults. To test this hypothesis, the effects of prenatal Cd on the development of autoimmune diabetes and arthritis were investigated. Non-obese diabetic (NOD) mice were exposed to Cd in a manner identical to our previous studies, and the onset of diabetes was assessed in the offspring. Our results showed a similar time-to-onset and severity of disease to historical data, and there were no statistical differences between Cd-exposed and control offspring. Numerous other immune parameters were measured and none of these parameters showed biologically-relevant differences between Cd-exposed and control animals. To test whether prenatal Cd-exposure affected development of autoimmune arthritis, we used SKG mice. While the levels of arthritis were similar between Cd-exposed and control offspring of both sexes, the pathology of arthritis determined by micro-computed tomography (μCT) between Cd-exposed and control animals, showed some statistically different values, especially in the female offspring. However, the differences were small and thus, the biological significance of these changes is open to speculation. Overall, based on the results from two autoimmune models, we conclude that prenatal exposure to Cd did not lead to a measurable propensity to develop autoimmune disease later in life.

## Introduction

Cadmium (Cd) is a heavy metal and carcinogen present in high levels at battery manufacturing facilities, around zinc smelters, and in cigarette smoke. Cd has been classified as a carcinogen

**Funding:** JBB ES023845. ES023845 National Institute for Environmental Health Sciences www. niehs.nih.gov The funders had no role in study design, data collection and analysis, decision to publish, or preparation of the manuscript. West Virginia University Flow Cytometry & Single Cell Core Facility OD016165 National Institute of General Medical Sciences https://www.nigms.nih. gov/ The funders had no role in study design, data collection and analysis, decision to publish, or preparation of the manuscript. Institutional Development Awards (IDeA) GM103488 (CoBRE) and GM103434 (INBRE) https://www.nigms.nih. gov/ The funders had no role in study design, data collection and analysis, decision to publish, or preparation of the manuscript. West Virginia University Animal Models & Imaging Facility RR016440 and P30 RR032138/GM103488 National Institute of General Medical Sciences https://www.nigms.nih.gov/ The funders had no role in study design, data collection and analysis, decision to publish, or preparation of the manuscript. West Virginia University Cancer Institute https://wvucancer.org/ The funders had no role in study design, data collection and analysis, decision to publish, or preparation of the manuscript.

**Competing interests:** The authors have declared that no competing interests exist.

based on epidemiological studies showing a link between Cd exposures with lung and breast cancers [1]. Humans absorb Cd either by ingestion or inhalation [2]. Recent studies on the urinary levels assayed as part of the NHANES reported overall Cd levels of 0.24 μg/g; however, levels of 0.41 μg/g were reported in smokers [3]. Urine Cd levels dropped by 34.3% in the interval between 1988–94 and 2003–2008, which is likely attributable to reduction in smoking [3]. The biological half-life of urinary Cd in the human body using a two-compartment model was 75–128 days for the fast compartment and 7.4–16 years for the slow compartment [4]. Additional studies using a larger cohort and a one-compartment model showed the urinary Cd half-life is around 12 years because of lack of effective Cd elimination pathways [5]. One of the organs with more Cd-induced toxicity is the thymus, the primary site of T-cell production [6]. Pharmacokinetic studies have shown that Cd does not readily reach the fetus in humans, but it accumulates in high concentrations in the placenta [7]. In humans, long-term Cd exposure *in utero* correlates with hypermethylation of genes in offspring cord-blood when compared to maternal leukocytes [8]. Additionally, it was recently reported that chronic exposure to Cd induces differential methylation in mice spermatozoa [9]. Taken together, these data indicate that there is likely an epigenetic mechanism by which the effects of Cd exposure can be transmitted to the offspring.

Previous studies from our group demonstrate that exposure of adult mice to environmental levels of Cd alters the immune cell development and function of their offspring [10–12]. In these studies, prenatal Cd exposure increased the number of CD4$^+$ cells and a subpopulation of double-negative cells (DN; CD4$^-$CD8$^-$), DN4 (CD44$^+$CD25$^-$) in the thymus at birth. These observations correlated with decreases in gene expression of both sonic hedgehog and Wnt/β-catenin [10]. Additionally, we assessed the ability of the 7-week-old offspring of Cd-exposed mice to respond to immunization with a heat-killed *Streptococcus pneumoniae* (HKSP). HKSP has two immunodominant antigens, pneumococcal surface protein A (PspA), which is a T-dependent antigen, and phosphorylcholine (PC), which is a type 2 T-independent antigen [13,14]. These studies revealed that prenatal Cd-exposure led to a >600-fold average increase in the IgG anti-PspA levels in females and about a 17-fold increase in males [11]. Anti-PC antibody levels also increased in these animals but in a less dramatic manner [11]. In contrast, splenic natural regulatory T cells (nTreg) were decreased in the female offspring [11]. Male offspring did not show a reduction in splenic nTreg cells [11]. At 20 weeks-of-age, offspring of Cd-exposed mice also exhibited persistent changes to thymus and spleen cell phenotypic repertoire, including decreased splenic nTreg cells in both sexes, as well as the adaptive immune response against HKSP [12].

The potential role of Cd in type 1 diabetes has been demonstrated in several experimental and epidemiological studies. Both acute and sub-chronic exposures to Cd have been shown to induce diabetogenic effects in animal models [15–17]. Additionally, Cd exposure during pregnancy correlates with an increased risk of gestational diabetes [18]. However, these studies assessed diabetes in humans and animals that were directly exposed to Cd. Given the reported direct effects of Cd on diabetes, increases in IgG antibody to HKSP, and the decreased nTreg cells in prenatally exposed females [11], we formulated the hypothesis that prenatal Cd may provide the initial immune cell changes that would lead to the development of clinical autoimmune disease as adults.

The foundation of our hypothesis was the reported role of nTreg and inducible regulatory T cells (iTreg) in autoimmune disease [19–25]. It is also apparent that the role of Treg cells in autoimmune disease is not necessarily correlated with just reduced numbers of Treg cells and that there are likely phenotypic differences, possibly caused by epigenetic changes [25–28]. However, to perform mechanistic studies on the effects of prenatal Cd on autoimmunity, whether due to changes in Treg cell numbers or function, requires an animal model in which

an outcome can be reproducibly attributed to the treatment. Thus, we sought to develop a model of Cd-induced autoimmunity by investigating the effects of prenatal Cd exposure using two well-established autoimmune animal models: the non-obese diabetic (NOD) mouse and the SKG Balb/C arthritis model, more recently established by Sakaguchi and coworkers [29]. Our models suggest that prenatal Cd exposure does not have a strong *in utero* effect on the development of autoimmune diseases in the offspring. This report provides guidance for other investigators in the pursuit of the goal of investigating the role of environmental exposures on autoimmune disease.

## Materials and methods

### Breeding and Cd exposure

All animal procedures were approved by the WVU Institutional Animal Care and Use Committee. Animals were euthanized by cervical dislocation as approved by the committee. Mice are housed in a specific pathogen-free (SPF) barrier facility with a 12 h light/dark cycle. Mice are fed normal chow (Envigo, 2018 Tekland Rodent Diet) and are allowed water ad libitum.

Non-obese diabetic (NOD/ShiLtJ; 001976) mice at 8–10 weeks of age were obtained from Jackson Labs (Bar Harbor, ME). Briefly, after a one-week (minimum) acclimation in our vivarium, 2 females were placed in a cage with one male for 5 days to maximize pregnancy rate. Dams used as controls had free access to normal water and Cd-treated dams had free access to 10 ppm of $CdCl_2$ (Sigma-Aldrich, C3141; $\geq$ 98% purity) dissolved in water. Briefly, $CdCl_2$ was reconstituted at 101.6 mg/L (5000X) in $ddH_2O$. Stock $CdCl_2$ was diluted to 10 ppm in $ddH_2O$, mixed thoroughly, and transferred to water bottles prior to autoclaving. To reduce the number of animals used for the study, only one dose was used at a level that would reproduce human environmental exposures, as described in the Introduction. The dose of 10 ppm was based on our previous studies which demonstrated immunomodulatory effects in the offspring of mice exposed to Cd [10–12]. Previous assessment of water consumption with or without Cd-spike, indicated no difference in water intake between groups [10]. Cd administration was stopped at birth. All offspring were weaned at 3–4 weeks and the dams were euthanized by $CO_2$ asphyxiation. Animals tested for the results reported herein were treated in an identical manner as a cohort tested at birth. In a previous cohort, Cd levels were assessed in the dams and representative offspring at birth using inductively coupled plasma optical emission spectrometry (ICP-OES) as reported in [10]. Cd levels in the kidneys of dams at the time of weaning the litter was 4.37±0.76 μg/g tissue and Cd levels were slightly above the level of detection by the ICP-OES (2.5 ppb) in the liver of offspring when tissues from ≥3 animals were pooled [10]. In this study, offspring at 18–22 weeks of age were euthanized by cervical dislocation and spleens were removed. The 18-week time point would approximate a 25-year-old human with a fully developed immune system. Mice were exposed to Cd in independent cohorts separated by time. Data are presented as combined data.

"SKG mice," discovered by Sakaguchi et al., are Balb/C mice that have a natural mutation in the *Zap70* gene encoding a component of the T cell receptor [30,31]. Raised under SPF conditions, these animals do not spontaneously develop rheumatoid arthritis (RA), but RA can be induced, as described below, and thus, a breeding colony can be maintained. SKG breeding pairs (a generous gift from Dr. Holly Rosenzweig, Oregon Health and Science University) were started with access to water with or without cadmium (10 ppm). Males were removed on day 5 and females were continued on cadmium-spiked or unspiked (control) water until pups were born. Cadmium bottles were exchanged with regular, unspiked water within 12 hours of the birth of pups. Offspring were weaned at day 28 postnatal. Two experimental cohorts were separated by time. Arthritic indices data are presented as summarized data.

## Autoimmune assays

Onset of diabetes in the NOD mice was determined by measuring urine glucose levels using the Keto-Diastix (Bayer USA, Whippany, NJ) urine analysis test strips. Weekly measurements were performed starting at 5 weeks and continuing through 18 (females) or 22 (males) weeks and the time-to-disease was noted for each animal.

RA was induced in the SKG SPF mice by intraperitoneal (IP) injection of 2 mg zymosan A (Sigma-Aldrich, St Louis MO; Z4250). These animals develop arthritis within 2–3 weeks of the zymosan A injection. Animals develop symptoms typical of arthritis, which includes swollen joints. An arthritic index (AI) is determined by scoring each paw individually on a scale of 0–4: 0 –normal; 1 –mild, but definite redness and swelling limited to individual digits; 2 –two swollen joints/digits; 3 –three or more swollen joints/digits; 4 –severe swelling of all joints/digits. All paws were evaluated, so that the maximal AI per mouse was 16. To alleviate pain, arthritic mice were given moist chow on a petri dish on the cage floor. Mice with severe arthritis were given soft bedding.

## Micro-CT

Following euthanasia, skeletons were fixed overnight in 10% formalin and stored in 70% ethanol at 4˚C until needed. Micro-computed tomography (µCT) was performed on a SkyScan 1272 Micro-CT (Bruker, Billerica MA). Images were processed and analyzed in accordance to standard methods [32]. Briefly, the left rear limbs of each animal were scanned in 70% ethanol using the following parameters: isometric voxel size 7 µm, tube voltage (x-ray tube potential) 60 kV, tube current 166 µA, exposure time 650 ms, aluminum filter 0.25mm. The cross-section images were reconstructed using NRecon (v 1.7.1.0; Bruker) with a 20% correction to reduce beam hardening and ring artifacts applied where appropriate. Segmentation was performed using local adaptive thresholding of the trabecular bone to reduce threshold biasing. The region of interest (ROI) was defined in CT Analyzer (v 1.16.9.0+, Bruker). Using the distal epiphysis (growth plate) as a reference point, the ROI was offset 245 µm (35 slices) proximal to ensure that no growth plate was included in the analysis of the metaphyseal trabecular bone. A total of 1.05 mm (150 slices) was used for quantitative analyses. Trabeculae were delineated from cortical bone using a manual assessment every 15–20 slices. Bone mineral density (BMD) was measured at the distal femur using the same Volume of Interest (1.05 mm) following calibration with calcium hydroxyapatite phantoms of known density: 0.25 and 0.75 g/cm$^3$. Phantoms were prepared, scanned, and reconstructed with the same parameters as above. Three dimensional images were prepared using DataViewer (v 1.5.6.2, Bruker) and CTVOX (3.3.0r1403, Bruker). Images representing the median mouse of each group are presented.

## Tissue isolation and cell preparation

Spleens were harvested from euthanized mice and single cell suspensions were prepared: spleens were dissected immediately after euthanasia and submerged into 5 ml of ice-cold phosphate-buffered saline without calcium or magnesium (PBS; Corning, 21-031-CV). Spleens were smashed between frosted sections of sterile microscope slides and passed through a 20G needle 3–4 times to obtain single-cell suspensions. Cells were washed once with PBS and red blood cells were lysed using RBC lysis buffer (Sigma-Aldrich, R7757). After neutralization with complete medium [RPMI-1640 medium (Corning, 15-040-CV) supplemented with 10% heat-inactivated fetal bovine serum (FBS, Sigma-Aldrich, F0926), L-glutamine (2 mM; Gibco, 25030–081), HEPES (5mM; Gibco, 15630–080), penicillin/streptomycin (100 units/L and 100 µg/ml; Cellgro, 30-002-C1), and 2-mercaptoethanol (0.05 mM, Sigma-Aldrich, M3148)], cells were washed once in staining buffer consisting of PBS pH 7.5, 0.5% bovine serum

albumin (BSA; Sigma-Aldrich, A7030), and 2 mM ethylenediaminetetraacetic acid (EDTA; Sigma-Aldrich, E5134) for flow cytometry staining or complete medium for cell culture stimulation. Viable cells were enumerated using trypan blue and a hemocytometer. Cell preparations from each mouse were analyzed individually for cytokines and splenocyte populations (see below).

## Primary cell culture and anti-CD3/28 stimulation

Splenocytes were stimulated with anti-CD3 and anti-CD28 in vitro. Briefly, 96 well tissue culture plates were coated with 50 μL PBS (with calcium and magnesium, Corning, 21-030-CV) or 5 μg/mL anti-CD3 (clone 17A2, eBioscience, 14-0032-86) diluted in PBS. Plates were incubated ~3 hours at 37˚C. Wells were washed twice with 200 μL PBS, cells were diluted to $1x10^6$/mL, and 200 μL of cells were placed in 5 wells per mouse per treatment (5 PBS, 5 anti-CD3). Cells were stimulated by the addition of anti-CD28 (clone 37.51, eBioscience, 14-0281-86) at 2 μg/mL final concentration and incubated for 72 hours at 37˚C with 5% $CO_2$. Supernatants were collected and stored at -80˚C until assay. Cell supernatants from each mouse were analyzed individually.

## Cell staining and flow cytometry

Single cell suspensions of splenocytes were prepared as described above. Surface and intracellular staining was performed as follows. Splenocytes were stained with either a cocktail of antibodies against CD3, CD19, CD4, CD8 (staining set 1) or CD4, CD25, and FOXP3 (staining set 2). All antibodies were used at a 1:200 dilution. A detailed list of staining panels, including antibody clones, fluorochrome tags, and the source of each antibody is provided in Table 1. Briefly, cells ($1 \times 10^6$) were washed with staining buffer and incubated with whole rat and mouse IgG (Jackson ImmunoResearch, West Grove, PA) for 30 min on ice to block Fc receptors. Without washing, cells were stained with LIVE/DEAD Fixable Yellow Dead Stain kit (Invitrogen, L34959) at a dilution of 1:2000 for 30 minutes on ice in the dark. Cells in both staining sets were washed once in staining buffer, incubated for 30 min on ice with fluorochrome-labeled antibodies to detect surface markers followed by several washes with staining buffer. For staining set 1, stained cells were fixed overnight at 4˚C with 0.4% paraformaldehyde. Paraformaldehyde was removed by washing with PBS and final resuspension in PBS. Fixed cells were stored at 4˚C until analyzed. For intracellular staining (staining set 2), cells were fixed and permeabilized using the eBioscience™ FOXP3/Transcription Factor Staining Buffer Set (Invitrogen, 00-5523-00) and stained with anti-FOXP3 antibody according to the manufacturer's protocol.

Data were acquired using a BD LSRFortessa and Diva 8.0 software (BD Biosciences) and analyzed using FCS Express software (De Novo Software, San Jose, CA). A minimum of 10,000 events were collected for each sample. Gating strategies are shown in S1 Fig.

## Cytokine analysis

Cell culture supernatants from the anti-CD3/CD28 stimulated splenocytes were analyzed for cytokine concentrations using MSD kit U-Plex Biomarker Group 1 (ms) Assays (Meso Scale Diagnostic, K15069L-1) with 9 analytes (GM-CSF, IL-17A, IL-17F, IL-1β, IL-21, IL-23, IL-6, KC/GRO, and TNF-α) according to the manufacturer's protocol. Samples were diluted 1:2 prior to addition to the plate. Plates were read on the MSD HTS24 plate reader.

**Table 1. Antibodies used for spleen phenotyping.**

Staining Panel 1

| Target | Clone | Fluorochrome | Manufacturer | Catalog # |
|---|---|---|---|---|
| CD3 | 145-2C11 | FITC | BD Pharmingen | 553062 |
| CD4 | RM4-5 | APC-eFluor 780 | Invitrogen | 47-0042-82 |
| CD8 | 53–6.7 | V450 | BD Horizon | 560489 |
| CD19 | 1D3 | PE | BD Pharmingen | 557399 |

Staining Panel 2

| Target | Clone | Fluorochrome | Manufacturer | Catalog # |
|---|---|---|---|---|
| CD4 (surface) | RM4-5 | APC-eFluor 780 | Invitrogen | 47-0042-82 |
| CD25 (surface) | 7D4 | PE | BD Pharmingen | 558642 |
| FOXP3 (intracellular) | FJK-16S | Efluor450 | eBioscience | 48-5773-82 |

### Anti-nuclear antibody screening

Sera from SKG mice were assessed for the presence of anti-nuclear antibodies (ANA) using the ANA Hep Screen ELISA Kit (Abnova, KA1080, lot 2106180). This assay is a screening system to qualitatively measure IgG class autoantibodies against SS-A-52 (Ro-52), SS-A-60 (Ro-60), SS-B (La), RNP/Sm, RNP-70, RNP-A, RNP-C, Sm-bb, Sm-D, Sm-E, Sm-F, Sm-G, Scl-70, Jo-1, dsDNA, ssDNA, polynucleosomes, mononucleosomes, histone complex, histone H1, histone H2A, histone 3, histone H4, Pm-Scl-100, and centromere B. Methodology was adapted to rodent studies by replacing the secondary antibody with HRP-conjugated goat anti-mouse IgG (BD Phamingen, Cat 554002, lot 5247553) in which is consistent with previous publications using similar technologies [33].

### Statistical analyses

All statistics were performed using GraphPad Prism version 8. Time to disease curves were compared within each sex using a Log-rank (Mantel-Cox) test and diabetes incidence was analyzed using Fisher's Exact test. When comparing two groups (*e.g.* Cd versus Control offspring), data were analyzed using two-tailed student's t-tests. When analyzing four groups (*e.g.* stratified by disease incidence), data were analyzed using an ANOVA with Tukey's post-hoc test for multiple comparisons. * $p < 0.05$, ** $p < 0.01$, *** $p < 0.001$. Raw data used to generate graphs presented as means are included in S1 Table.

## Results

### Incidence of diabetes in NOD offspring

To determine the effect of prenatal Cd exposure on autoimmune disease, the incidence and time-to-disease of diabetes was measured in NOD offspring. NOD parents were exposed to Cd as described under Methods and the offspring monitored for elevated glucose as an indicator of diabetes. Female Cd offspring showed a slight delay in the development of diabetes as compared to female control offspring (Fig 1A). This difference was not statistically significant. The percentage of diabetes-positive female offspring and the number of litters composing each group is shown in Fig 1B. Male offspring developed diabetes (both treatment groups) later than the female offspring (Fig 1A). There was essentially no difference in the incidence of diabetes or time of onset between the male Cd offspring and control offspring (Fig 1A and 1B). Thus, by these measures prenatal exposure to Cd did not cause a significant change in autoimmune disease development.

A

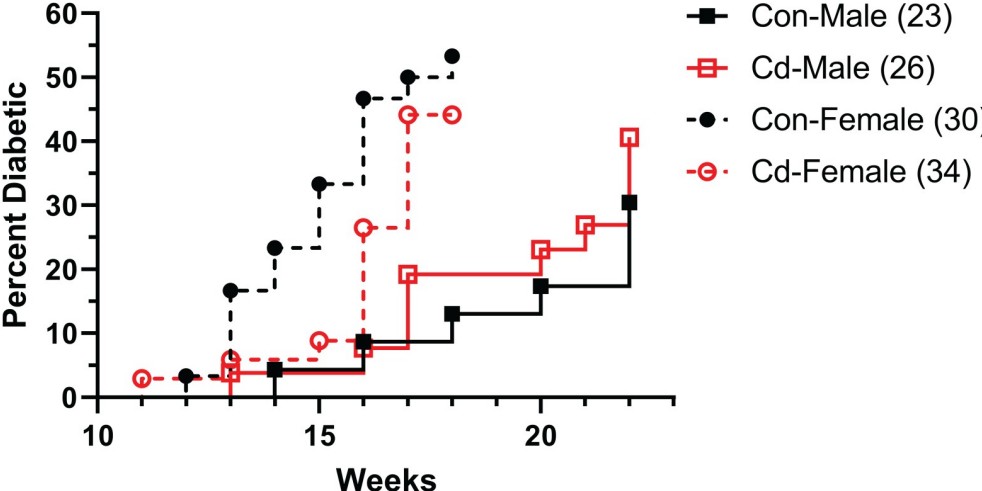

B

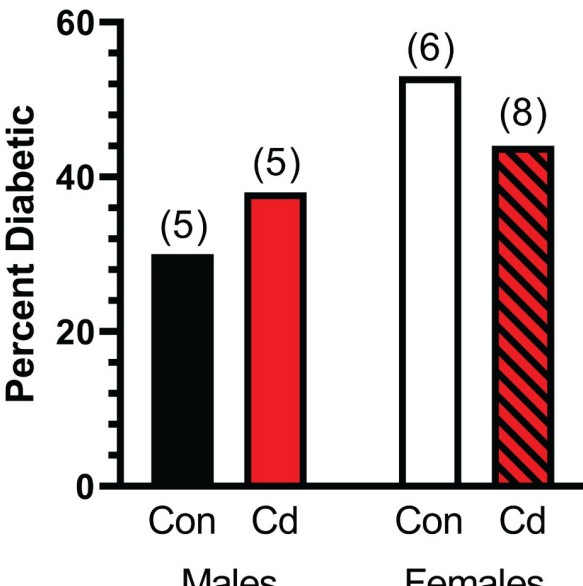

**Fig 1. Time to disease and incidence of diabetes of Cd-exposed NOD offspring.** Diabetes incidence was based on elevated glucose in the urine of the offspring at the indicated times. **(A)** Time (weeks) until a positive urine test. Number of mice per group are indicated in the legend. **(B)** Percent of offspring in the litters that were diabetic at 18 weeks (females) and 22 weeks (males). Number of dams associated with these litters shown above the bar.

## Splenic phenotypes in NOD offspring

Splenic phenotypes were assayed in the offspring at 18 (females) and 22 (males) weeks of age. S2 Fig (S2 Fig) shows the number of CD4[+], CD8[+] and CD19[+] splenocytes in each group. No differences in the numbers of any of these cell phenotypes were seen.

We previously reported decreased numbers of nTreg cells in the Cd-exposed offspring in a C57BL/6 mouse model [11]. Thus, we sought to determine if nTreg cell numbers in the NOD offspring was also altered by prenatal Cd exposure. We first assessed whether prenatal Cd exposure altered Treg percentages, regardless of disease state. As shown in Fig 2, prenatal Cd exposure does not alter nTreg percentages. We hypothesized that offspring with measurable disease would have reduced Tregs in the spleen. However, no differences in nTreg cell number were observed when stratified by disease state in any treatment group of either sex (Fig 3). No changes in total splenocyte numbers were observed between any of the treatment groups for either sex. Therefore, Cd exposure did not cause changes in the numbers of the total cell number, major splenocyte phenotypes, or specifically of the Treg cells.

## Splenic cytokine production in NOD offspring

Several cytokines are associated with the development of autoimmune disease [34,35]. These are predominantly the cytokines that contribute to inflammation, and include IL-17A and IL-17F as well as TNF-α. In male offspring, at 22 weeks there was a statistically significant decrease in IL-17F between the control diabetic and the Cd-exposed diabetic and non-diabetic animals (Fig 4); however, whether these differences were biologically significant is not obvious. No differences were seen in either IL-17A or TNF-α levels in these male offspring.

Similar assays were performed on the female offspring at 18 weeks. IL-17A was reduced in non-diabetic female Cd animals as compared to non-diabetic controls (<2-fold, $p<0.05$), indicating a possible inhibiton of inflammation with prenatal Cd exposure in the female offspring (Figs 4 and S3). Again, the biological significance of this difference is not obvious (Fig 4). No significant differences were noted in the cytokine production levels of any of the other cytokines shown in Fig 4.

Additional cytokines were also measured in the spleen cells from these offspring. These included IL-1β, IL-21, IL-23, IL-6 and KC/GRO (S3 Fig). No significant differences were noted in the ability of splenocytes to produce these cytokines were seen in either sex regardless of treatment group.

## Arthritis incidence in the SKG mouse strain

The specific-pathogen-free (SPF) reared SKG mice develop arthritis after zymosan A injection. At 8 weeks, arthritis was induced in male and female SKG offspring to determine if differences in the arthritis incidence or severity is due to prenatal Cd exposure. Arthritic index (AI), which is a measure of arthritic-inflammation in the paws was assessed weekly starting one week post zymosan A-induction. A positive AI score (as further described in the Methods) is indicated by visible swelling in one or more joint within the paw. Each paw is measured on a scale of 1–4 with a total possible score of 16 per animal (4 paws). No differences in AI between the treatment groups in either sex was apparent (Fig 5). However, females, regardless of treatment, showed a positive AI 3 to 4 weeks earlier than males. By this measure, prenatal Cd exposure did not cause an increase in arthritic inflammation.

Direct Cd exposure via drinking water has been shown to induce the presence of antinuclear antibodies (ANA) in the blood of mice [33]. To assess the presence of ANA in the

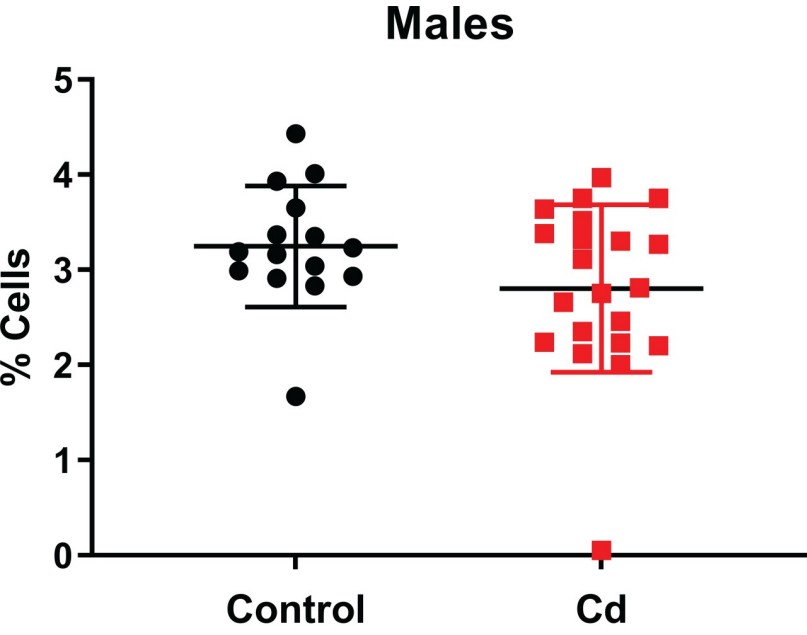

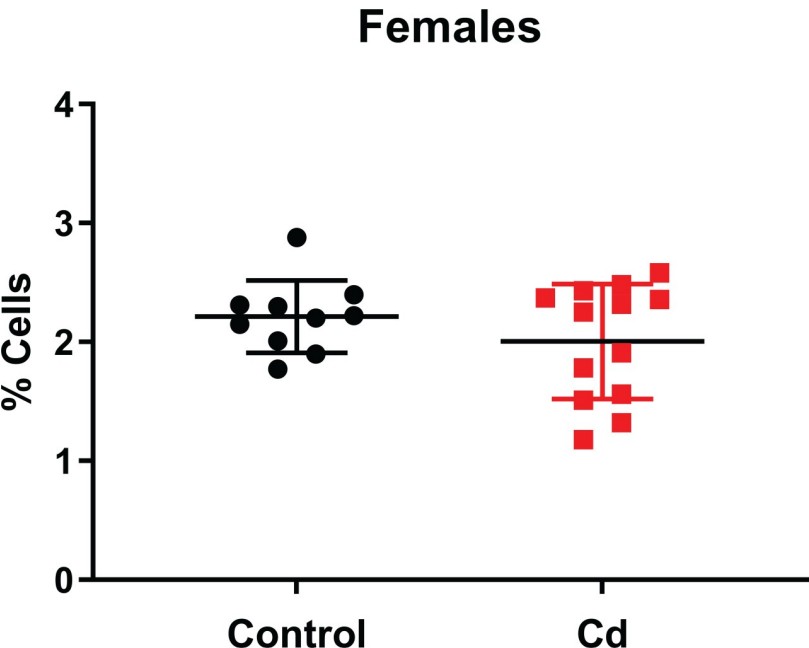

**Fig 2. Analysis for Treg spleen cell percentages of Cd-exposed NOD offspring.** At 18 weeks (females) and 22 weeks (males), the spleens of the offspring were ablated and processed for flow cytometric analysis. CD4$^+$CD25$^+$FOXP3$^+$ were quantified. Data are shown as mean ± SD. Males N = 15–21; Females N = 10–13.

offspring of exposed mice we screened for autoantibodies in the sera using an ELISA method adapted for rodents. A single male mouse exhibited autoantibodies above background (S4 Fig); however this did not correlate with increased disease, i.e. arthritic index.

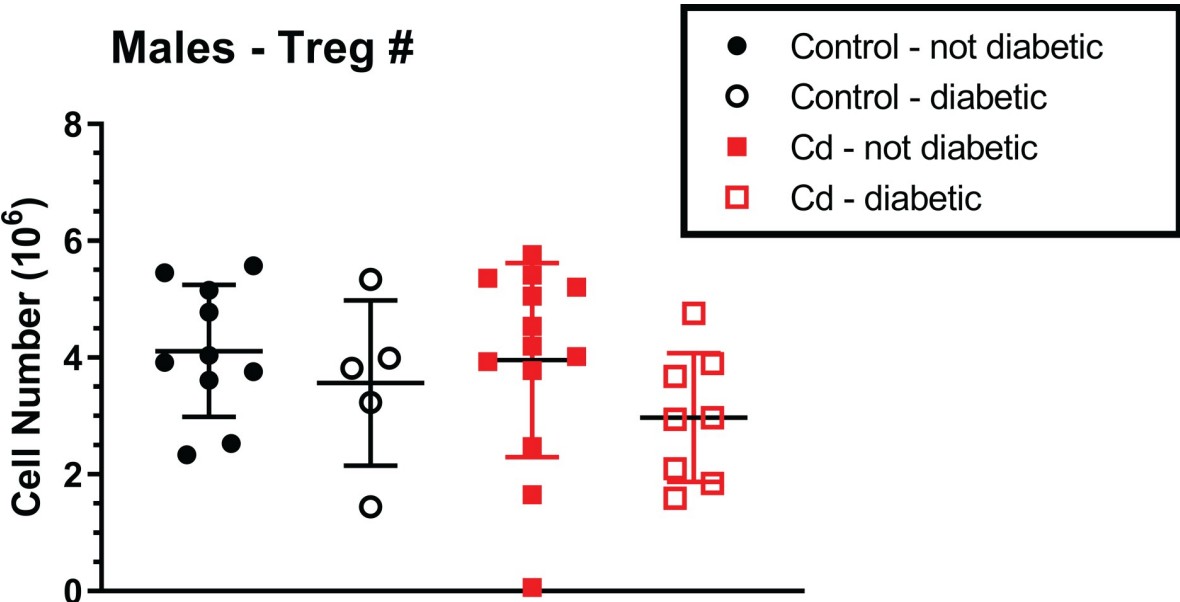

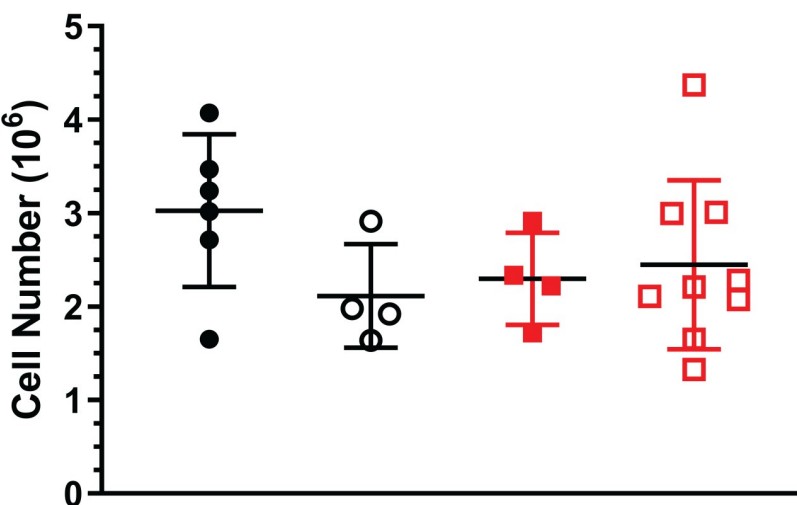

**Fig 3. Analysis for Treg spleen cells of Cd-exposed NOD offspring.** At 18 weeks (females) and 22 weeks (males), the spleens of the offspring were ablated and processed for flow cytometric analysis. CD4+CD25+FOXP3+ were quantified. Data are shown as mean ± SD. Males N = 5–13; Females N = 4–9.

### Microcomputed tomographic analysis of the femurs of arthritic SKG mice

In our experience, the AI does not accurately reflect possible bone erosion that accompanies arthritis. Micro-CT (μCT) has been considered the gold standard for assessing trabecular bone microarchitecture [36,37]. For this reason, μCT visualization and analyses were also conducted on the paws of these SKG mice. Fig 6 shows a typical sample from representative animals of all experimental groups. As noted in Fig 6B and 6D, two representatives from the male and female

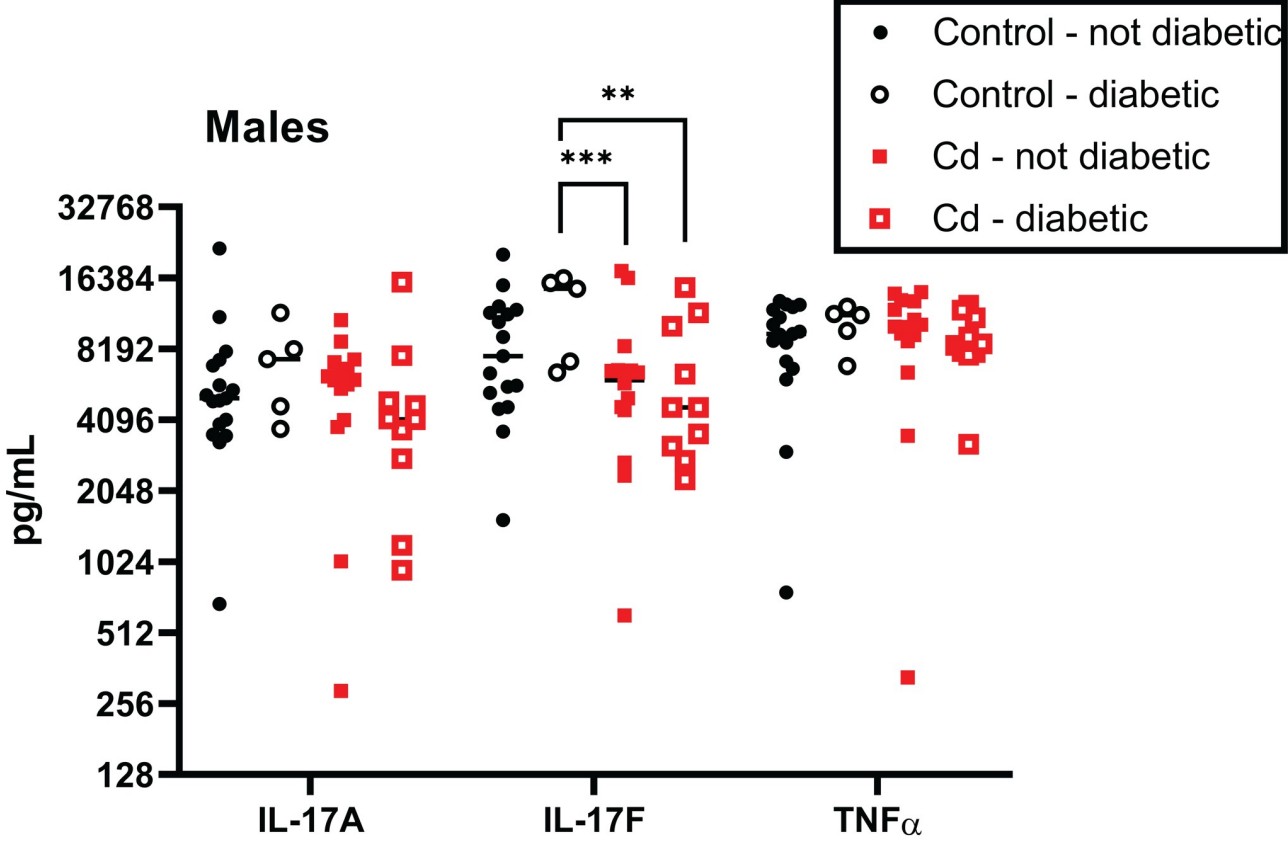

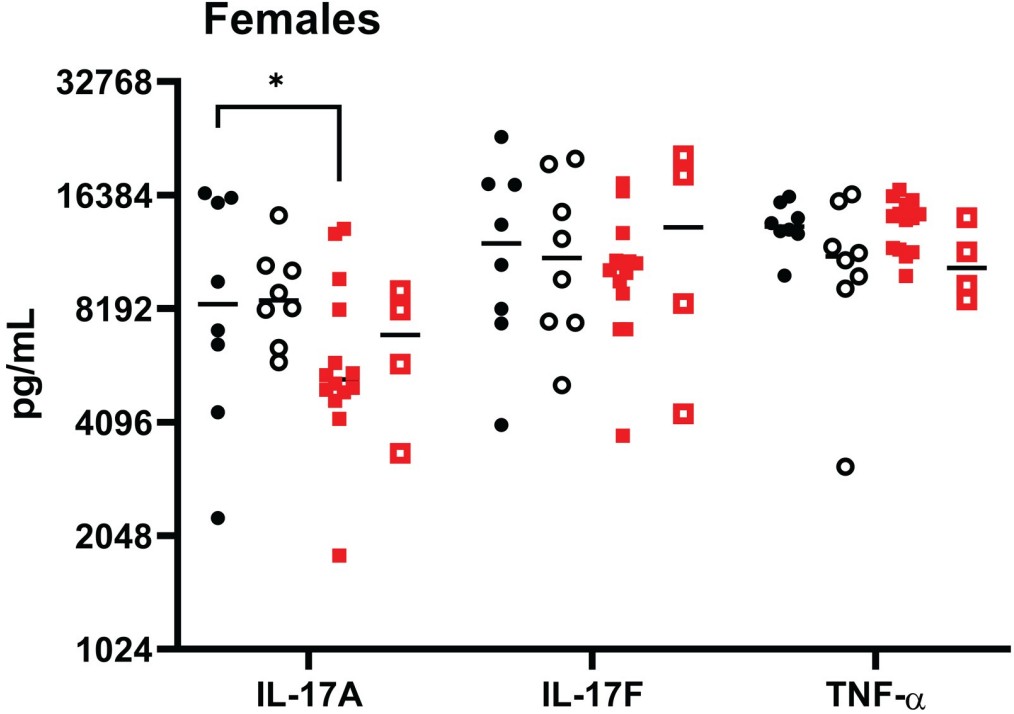

**Fig 4. Spleen cell proinflammatory cytokine production by Cd-exposed NOD offspring.** At 18 weeks (females) and 22 weeks (males), the spleens of the offspring were ablated and stimulated with anti-CD3+anti-CD28 for 72h. Cytokine levels were measured using MSD multiplex plates. Symbols represent the data for individual mice and the black bar denotes the median. Males N = 5–17; Females N = 4–14.

cohorts displayed essentially identical AI scores. Visible in the x-ray scan of the Cd female (Fig 6A) is an obvious distortion (indicated by an arrow) in the phalanges of the paw. This distortion is not present in the control female; however, it is not clear whether this was due to increased brittleness and distortion caused by in vitro experimental manipulation or represents a true bone defect. This distortion was not seen consistently in all Cd female μCT scans. No other differences in the μCT x-ray scans were noted in male offspring, of either treatment group.

Quantitative morphometry was performed on representative μCT images of the femurs from each experimental group and the results are shown in Fig 6. The volume of interest (VOI) was defined as described in Micro-CT methods. Representative images from each sex and treatment group are illustrated in Fig 7A, with the VOI highlighted in red. Several parameters were assessed: 1) percent bone volume (BV/TV) which quantifies the percent of space occupied by trabecular bone within the VOI, 2) trabecular number is also known as the spatial density and determined by the number of times per mm that random line in the VOI crosses a trabecula, 3) trabecular thickness is the average thickness as measured by virtual sphere-fitting, 4) trabecular separation is the diameter of spaces between the trabeculae as assessed using 3D sphere fitting, 5) bone surface to bone volume ratio (BS/BV) which is a general indicator of structural complexity and inversely related to trabecular thickness, and 6) bone surface to total

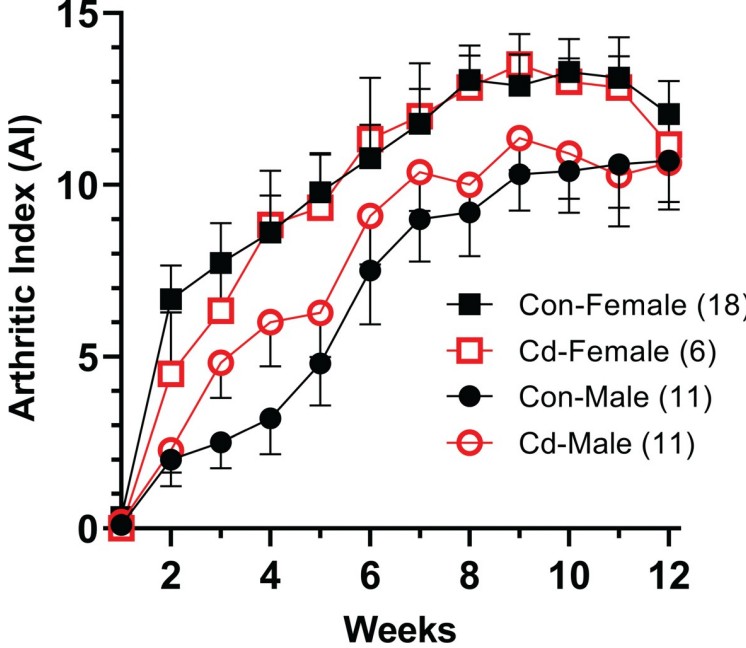

## Zymosan A Induction of Arthritis in SKG Mice

**Fig 5. Incidence of arthritis in Cd-exposed SKG offspring.** At 8 weeks, the arthritis was induced in the offspring of control and Cd-exposed parents by the injection of zymosan A. The arthritic index was determined in the over a 12-week period of time. Data from two independent cohorts were combined are presented as mean ± SEM. Number of mice per exposure group are indicated in the legend.

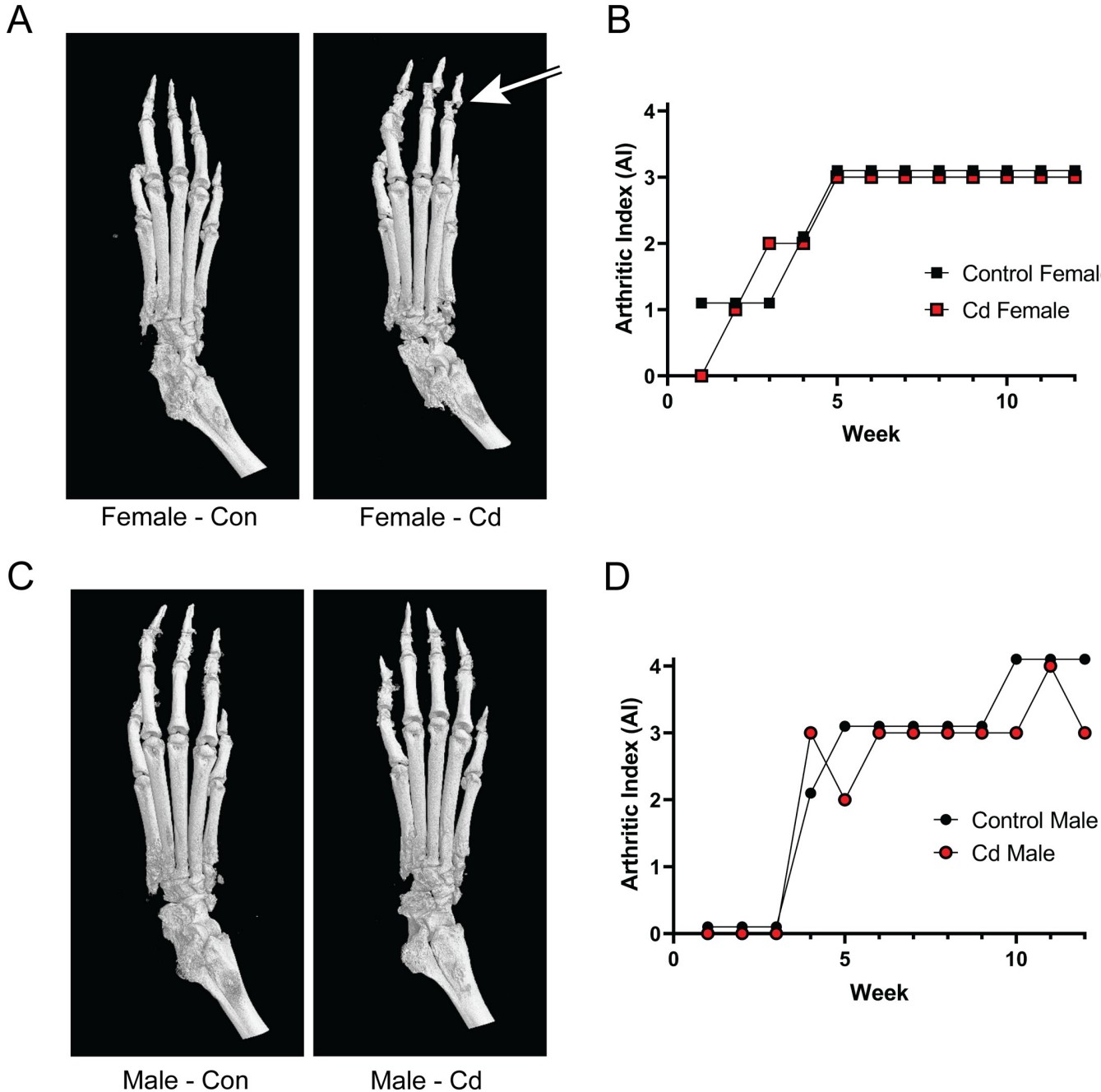

**Fig 6. MicroCT analysis of the paws of arthritic Control and Cd-exposed SKG offspring.** Dam and sire SKG mice were exposed to cadmium as described. At 8 weeks, the arthritis was induced in the offspring by the injection of zymosan A. Twelve weeks after arthritis induction, the animals were euthanized, and the limbs processed for μCT analysis. **(A & C)** Representative μCT images from rear left paws. Distorted phalanges are indicated by the arrow. **(B & D)** Arthritic indices for individual paws shown in A & C. No arthritis control values are inidicated by the dotted line.

VOI ratio (BS/TV) which quantifies the bone surface density. The Cd females showed a statistical difference in the bone volume fraction (BV/TV) and trabecular thickness (Fig 7B)–values that have been deemed crucial for determining the three-dimensional (3D) outcomes for trabecular bone microarchitecture [36]. The ratio of the segmented bone surface to the segmented bone volume (BS/BV; Fig 7) was also statistically different in the female Cd-exposed

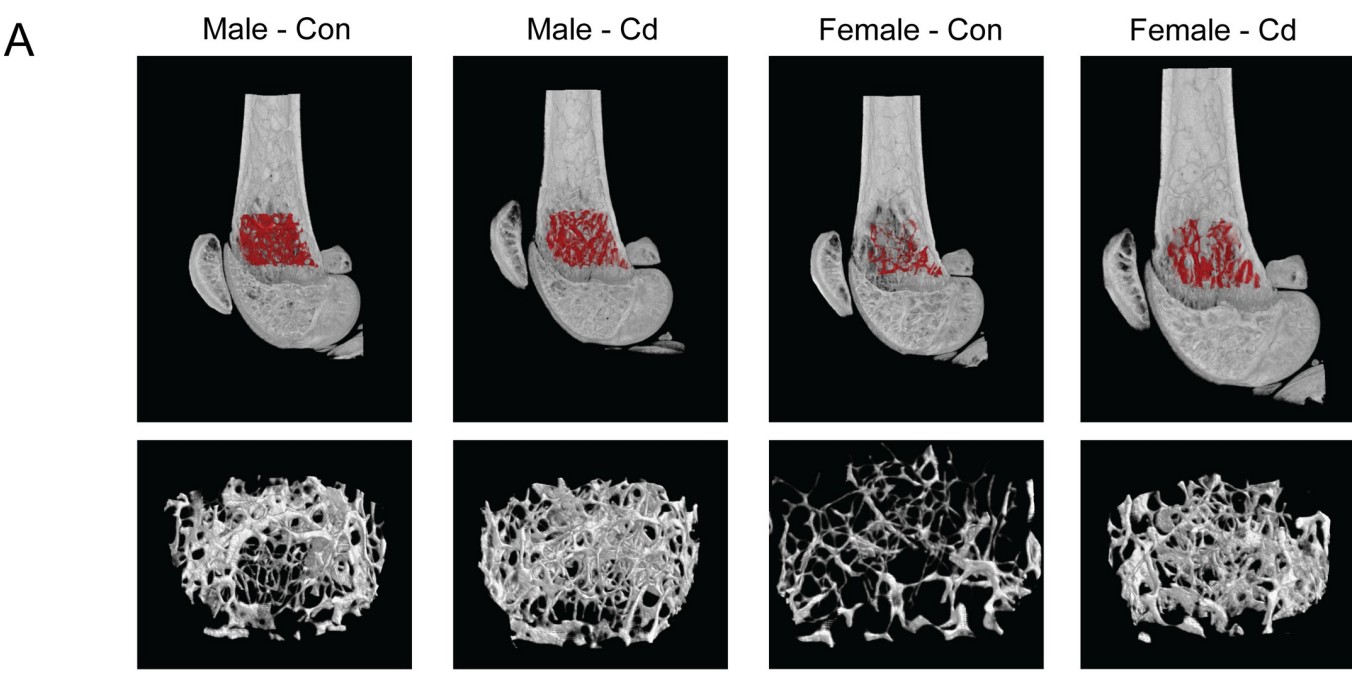

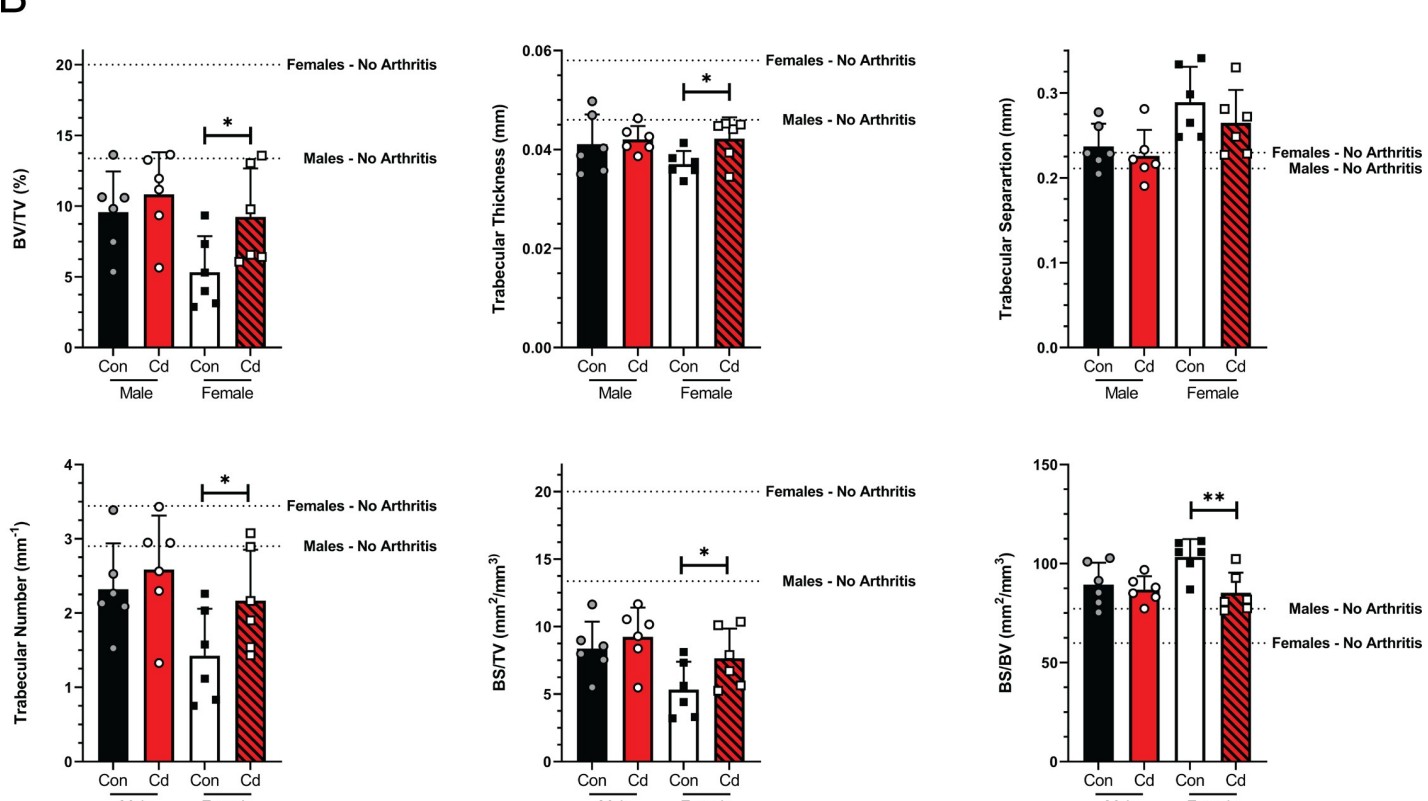

**Fig 7. MicroCT analysis of the femurs of arthritic Control and Cd-exposed SKG offspring.** Femurs of control and Cd-exposed offspring were analyzed by μCT. **(A)** Cross sections of μCT images of mice representing median mice presented in B. The VOI is highlighted red and shown enlarged below. **(B)** Quantification of trabecular bone architecture. BV/TV–bone volume/total volume of interest, BS/TV–bone surface/total volume of interest, BS/BS–bone surface/bone volume. Data are presented as mean ± SD. N = 4 per group.

offspring. Male offspring showed no statistical differences in any assessment of the trabecular bone architecture (Fig 7).

BMD was assessed in the femurs of male and female offspring following prenatal Cd exposure. Values for the no arthritis control mice are indicated by the dotted line. BMD was significantly increased in the femurs of Cd-exposed female offspring as compared to the controls (S6 Fig). No differences were observed in the male offspring.

## Discussion

Autoimmune disease is a topic of intense interest because of its life-changing consequences. Treg cells are widely regarded as the key to preventing autoimmune disease because of their well-established role of controlling immune responses and a plethora of studies that have established a potential role of Treg cells in autoimmunity development. In humans, one possible scenario is that some defining initiating event alters Treg cells such that subsequent event (s), presumably separated by time, precipitate alteration in numbers and/or functions of Tregs in a manner that results in autoimmunity. Often neither the initial defining event(s), nor the triggering event(s) are known. It is also too simplistic to attribute autoimmunity solely to the aberrant actions of the Treg cell populations. In addition, heterogeneity of responses by humans to the same stimuli makes these key events increasingly difficult to define.

Exposure to environmental xenobiotics has been postulated as a possible initiating event but may also later trigger frank autoimmune disease [33]. There are a wide range of environmental agents that have been suggested as possible triggers [38]. There are several indicators that Cd may be involved in developing susceptibility to autoimmune disease. For example, direct Cd exposure of adults has been implicated in diabetes development (reviewed in [39]). A relatively unexplored exposure paradigm is the possible epigenetic changes induced in the developing fetus when the parents are exposed to an environmental agent. In this scenario, the environmental agent would be considered the initiating event with the subsequent triggering event undefined. Our previous reports show that prenatal Cd exposure altered several key immune parameters in the offspring suggesting that this treatment could be the initiating event possibly leading to development of autoimmune disease. Based on these data, we embarked on studies to develop a reliable animal model that would allow us to study the role of Cd in prenatally-induced autoimmune susceptibility.

The first autoimmunity model that we tested was the NOD mouse model. NOD mice have been extensively used to investigate the mechanism of autoimmune-induced diabetes and given the previous reports of Cd inducing diabetes with direct exposure, it seemed like a reasonable model to explore. There are numerous reports linking the onset of diabetes in the NOD mouse to functional changes in Treg cells, which prompted us to hypothesize that prenatal exposure of NOD mice to Cd might lead to acceleration of diabetes as in other model systems [40]. Our findings on time-to-onset in NOD mice agree with published studies [40]. Based on the incidence of diabetes, females showed diabetic symptoms earlier and in a higher percentage than male counterparts. Within the female cohort, Cd-exposed offspring generally did not show signs of diabetes until approximate 2 weeks later than the control group; however, once they began to show diabetes, the Cd-exposed group quickly caught up with the controls (Fig 1A). Whether this delay in onset and more rapid progression of the Cd-exposed group has biological significance is unknown. However, we found no statistically significant differences in our studies. Males, in contrast, showed similar onset timing and incidence regardless of treatment group.

In this study we found that prenatal Cd exposure did not alter splenocyte cell populations; this is in contrast to our previous publication where prenatal Cd exposure was associated with

reduced splenic Tregs in female offspring and increased splenic Tregs in male offspring [11]. Of note, the previous study examined the effects in offspring using a C57BL/6 background. Prenatal Cd exposure may not induce the same changes in nTreg percentages and overall numbers in the NOD mouse model which has a NOD/ShiLtJ background. Although there is a large body of literature designating the defect in NOD mice as Treg mediated, there is also a report that the defect in NOD is in the T-effector (Teff) cell population [41]. Therefore, a lack of a change in diabetes in this mouse model, may not preclude a possible defect in the Treg population. The effect of prenatal Cd exposure on Treg suppressive function should be assessed in future studies.

To assess T cell function, we stimulated splenocytes *in vitro* with anti-CD3/anti-CD28 and measured proinflammatory cytokines secreted in the media (Figs 4 and S3). We predicted that prenatal Cd exposure would increase IL-17 levels in the sera. However, we observed that IL-17A was reduced in non-diabetic female Cd animals as compared to non-diabetic controls and IL-17F levels were lower in male Cd-exposed (diabetic and non-diabetic) offspring than in control diabetic animals. However, the data are very noisy and do not engender confidence that these differences have any biological significance, especially when IL-17 levels were decreased in the Cd-exposed offspring. Previous studies in the NOD model show that IL-17A and IL-17F expression correlates with increased autoimmune-mediated type I diabetes [42] and pharmacological inhibition of Th17 cells prevented the development of diabetes in these mice [43]. Conversely, IL-17 deficient NOD mice only showed delayed onset of diabetes and did not result in decreased cumulative incidence in this model [44]. Taken together, we would predict that diabetic mice in our study would have increased IL-17 levels as compared to the controls. However, we did not observe differences in IL-17 production between non-diabetic and diabetic mice regardless of prenatal exposure. Of note, we measured *ex vivo* cytokine production of splenocytes. These measurements may not be indicative of IL-17 production *in vivo* in organs important to the development and progression of type 1 diabetes, such as the pancreas. Based on these observations, we conclude that prenatal Cd exposure did not alter the development of diabetes in NOD mice.

In addition to IL-17, we assessed the levels of TNF-α, IL-1β, IL-21, IL-23, IL-6, and KC/GRO. No differences were observed in the ability of T cells to produce these cytokines upon stimulation. In our previous C57BL/6 studies [11,12], we reported significant reduction in IFNγ production in prenatal Cd offspring; however, we did not measure IFNγ levels in the present study. While IFNγ is generally thought to be important in the development of type 1 diabetes, in the NOD model, the lack of IFNγ or IFNγ receptor does not prevent the development of disease [45,46]. Furthermore, the role of IFNγ in animal models of RA remains controversial as it is shown to be protective in the collagen-induced model but required for arthritis development when induced by glucose-6-phosphate isomerase [47]. However, we acknowledge that a limitation of our study is that we did not assess IFNγ or anti-inflammatory cytokines in these models.

Given the failure to demonstrate that the NOD mouse was a suitable model to study possible prenatal Cd autoimmune induction, we tested a second autoimmune model which presents with arthritis. The SKG mouse arthritis model, identified by Sakaguchi and associates [29], has a mutation in the ZAP70 signaling molecule that was caused by a spontaneous event. The mutation alters developing T cells. Treg cell production and function are affected allowing the development of polyclonal self-reactive arthritogenic T cells [48]. These mice, when raised in SPF conditions typical in modern vivaria, remain disease-free until injected with zymosan A (this report) or mannan [49]. Once induced, animals develop arthritis within 2–3 weeks. This model was chosen over collagen-induced arthritis because the arthritis occurs in SKG females earlier and with greater severity than males, which recapitulates the demography of human

arthritis [49]. The Cd and control female offspring showed earlier and slightly more severe arthritis than their male counterparts; however, within a given sex, no statistic or apparent differences were observed (Fig 5).

Previous studies have demonstrated presence of ANA in 10% of female FZBW F1 mice exposed to 10 ppm Cd for 4 weeks [33]. To test if ANA were present in the offspring of mice exposed to Cd water for 21 days, we screened the plasma using an ELISA-based method for IgG class autoantibodies against SS-A-52 (Ro-52), SS-A-60 (Ro-60), SS-B (La), RNP/Sm, RNP-70, RNP-A, RNP-C, Sm-bb, Sm-D, Sm-E, Sm-F, Sm-G, Scl-70, Jo-1, dsDNA, ssDNA, polynucleosomes, mononucleosomes, histone complex, histone H1, histone H2A, histone 3, histone H4, Pm-Scl-100, and centromere B. In contrast to direct Cd exposure, only a single male offspring demonstrated levels above background (S4 Fig) in our study. Prenatal Cd exposure did not cause the reduction in spleen weight observed with the Leffel et al. study [33]. Taken together, these data suggest that prenatal Cd exposure does not elicit the same effects in the offspring as direct Cd exposure does in adult animals. Cd does not cross the placental barrier [7].; therefore, it is not suprising that direct Cd exposure may have a different mechanism of action than indirect exposure in the womb. Further experiments are needed to elucidate the mechanisms by which prenatal Cd exposure drives immunological changes in the offspring.

In other experimental systems used in our laboratory, we have noted that the arthritic index (Fig 5), does not necessarily reflect changes in bone architecture. We therefore employed µCT to investigate bone architecture in control animals and offspring of Cd-exposed animals (Figs 6 and 7). X-ray micrographs from representative animals (Fig 6) do not show any notable differences in the bone architecture of the paw beyond what was previously noted as deformation of the phalanges. Quantitative analysis of femurs (Fig 7) revealed small but statistically significant differences in several parameters within the female offspring cohort, including BV/TV, trabecular thickness, and BS/BV. While statically significant, the numerical difference in the mean trabecular bone thickness in the female offspring is only 5.2 µm which is smaller than one pixel (7 µm). Therefore, this difference is less than the resolution of our instrumentation and whether these differences are real or represent a true biological defect is doubtful. As the BS/BV measurement is inversely proportional to the trabecular thickness, the data presented are deceptive and may not accurately reflect the biology.

Unexpectedly, the percentage of bone volume as compared to the total volume (BV/TV) in our VOI was significantly higher in the prenatal Cd female offspring as compared to the control female offspring. The detrimental effects of Cd on bone development have been previously reported with both direct exposure to cells [50–52] as well as prenatal exposure in rats [53]. Low levels of Cd (7.5–60 nM) stimulate osteoclastic differentiation in the presence of osteoblasts [51]. Additionally, Cd induces apoptosis in primary rat [52] and human [50] osteoblasts at concentrations of 1–5 µM and 10–50 µM, respectively. Taken together, these studies demonstrate that direct Cd exposure both promotes bone destruction through osteoclast differentiation and inhibits bone repair through the destruction of osteoblasts.

Prenatal Cd exposure (50 ppm via drinking water) in rats results in decreased femur weight, but not body weight, and length [53]. The authors demonstrate that this was due to the ability of Cd to prevent transfer of essential elements, including zinc, to the fetus. The phenotype was rescued with supplemental zinc in diet of the pregnant dam [53]. In contrast, our data indicate that the female Cd offspring had increased BV/TV as compared to the female controls (albeit similar to male Cd and control offspring); therefore, we cannot reasonably conclude that prenatal Cd exposure is causing the differences observed in our study. In attempt to identify additional explanations for our data, we assessed the weight and limb-specific arthritis in the female offspring. The zymosan A injection which induces RA in this SKG model is a single dose of 2 mg, regardless of animal weight. We determined that the weights of the female

offspring at the time of the arthritis induction were not different (S5A Fig) and therefore did not result in the observed differences in bone structure. Further, we know that arthritis development in this mouse model, as well as the collagen-induced arthritis (CIA) model, is asymmetric in the limbs. We hypothesized that the differences observed in our trabecular microarchitecture measurements are purely due to varying degrees of arthritis in the scanned limb. However, we found no differences in AI score among the groups (S5B Fig). Finally, while statistically significant, the BV/TV percentage in female control and Cd offspring are highly variable and the difference in the means is small (5.8%). The true biological significance of this parameter is open to speculation. No differences in trabecular bone architecture were observed in the control and Cd male SKG offspring.

Patients with rheumatoid arthritis have a higher risk of developing low BMD and osteoporosis. BMD was assessed in the femurs of male and female offspring following prenatal Cd exposure. BMD was significantly increased in the femurs of Cd-exposed female offspring as compared to the controls (S6 Fig). Consistent with the microarchitecture measurements above, Cd-exposed female offspring had higher BMD than the control offspring. Of note, the control offspring appear to have more bone erosion and less BMD than any other group. If anything, taken together, these data suggest that prenatal Cd exposure appears to be protective in the development of arthritis in the female offspring using the SKG model. Therefore, we conclude that prenatal exposure to Cd does not predispose or exacerbate autoimmune disease in NOD and SKG mice.

## Conclusions

In summary, this report details our attempts to develop a reliable model to test the hypothesis that prenatal Cd exposure contributes to the development of autoimmunity. While there were tantalizing hints of an effect, neither of the two models investigated provided data that would unequivocally denote that prenatal Cd induced a higher propensity for Cd-exposed offspring to develop autoimmune disease. Without a consistent model, mechanistic experiments were not possible. However, the data reported herein highlight the importance of model validation and, we hope, will assist those using mouse models of autoimmunity to investigate the long-term outcomes of prenatal exposures.

## Supporting information

**S1 Fig. Gating strategies for flow cytometry panels.**
(TIF)

**S2 Fig. Spleen cell phenotypes of cadmium exposed NOD mice.** Dam and sire NOD mice were exposed to cadmium as described. At 18 weeks (females) and 22 weeks (males), the spleens were ablated and processed for flow cytometric analysis. CD4$^+$, CD8$^+$ and CD19$^+$ were quantified. Symbols represent the data for individual mice and the black bar denotes the mean. Males N = 6–16; Females N = 13–19.
(TIF)

**S3 Fig. Spleen cell cytokine production by cadmium exposed NOD mice.** Dam and sire NOD mice were exposed to cadmium as described. At 18 weeks (females) and 22 weeks (males), the spleens of the offspring were ablated and stimulated with anti-CD3+anti-CD28 for 72h. Cytokine levels were measured using MSD multiplex plates. No differences were noted in production of these cytokines between any group. Symbols represent the data for individual mice and the black bar denotes the median. Males N = 5–17; Females N = 4–14.
(TIF)

**S4 Fig. Prenatal Cd exposure does not induce ANA in SKG model.** Serum autoantibodies were measured by ANA Hep Screen ELISA.
(TIF)

**S5 Fig. SKG mice body weight at time of induction and AI of left rear paws.** A) Body weights of male and female mice at the time of arthritis induction with zymosan A. Symbols represent the data for individual mice and the black bar denotes the median. B) Left rear paw arthritic indices. No differences were noted between control and cadmium exposure within each sex. Data are shown as mean ± SEM.
(TIF)

**S6 Fig. BMD analysis of the femurs of arthritic Control and Cd-exposed SKG offspring.** Bone mineral density (BMD) was measured at the distal femur using the same Volume of Interest (1.05 mm) following calibration with calcium hydroxyapatite phantoms of known density: 0.25 and 0.75 g/cm$^3$. Phantoms were prepared, scanned, and reconstructed with the same parameters as Fig 7. Data are shown as mean ± SD.
(TIF)

**S1 Table. Raw data used for graphs presented as means.**
(XLSX)

## Acknowledgments

We thank Michelle Witt for her training in microCT image acquisition and analyses.

## Author Contributions

**Conceptualization:** John B. Barnett.

**Formal analysis:** Harry C. Blair, Casey Hall.

**Investigation:** Jamie L. McCall, Kathryn E. Blethen, Casey Hall.

**Methodology:** Meenal Elliott.

**Visualization:** Jamie L. McCall, Casey Hall.

**Writing – original draft:** John B. Barnett.

**Writing – review & editing:** Jamie L. McCall, Harry C. Blair, Meenal Elliott.

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
