## [Decision Letter · Decision Letter 0]

19 Apr 2021

PONE-D-21-08667

Prenatal cadmium exposure does not induce greater incidence or earlier onset of autoimmunity in the offspring

PLOS ONE

Dear Dr. Barnett,

Thank you for submitting your manuscript to PLOS ONE. After careful consideration, we feel that it has merit but does not fully meet PLOS ONE’s publication criteria as it currently stands. Therefore, we invite you to submit a revised version of the manuscript that addresses the points raised during the review process.

All three reviewers make important suggestions on how to improve the manuscript, ranging from content to presentation.  Please go carefully through all review comments below.

We look forward to receiving your revised manuscript.

Kind regards,

David M. Lehmann, Ph.D.

Academic Editor

PLOS ONE

Journal Requirements:

To comply with PLOS ONE submissions requirements, in your Methods section, please provide additional information on the animal research and ensure you have included details on methods of anesthesia and/or analgesia, and efforts to alleviate suffering.

We note that you have included the phrase “data not shown” in your manuscript. Unfortunately, this does not meet our data sharing requirements. PLOS does not permit references to inaccessible data. We require that authors provide all relevant data within the paper, Supporting Information files, or in an acceptable, public repository. Please add a citation to support this phrase or upload the data that corresponds with these findings to a stable repository (such as Figshare or Dryad) and provide and URLs, DOIs, or accession numbers that may be used to access these data. Or, if the data are not a core part of the research being presented in your study, we ask that you remove the phrase that refers to these data.

4. PLOS ONE requires that the data underlying each figure and table be made available to the public prior to publication (https://journals.plos.org/plosone/s/data-availability).  Please review the journal’s policy and provide a suitable mechanism to satisfy the requirement. [NOTE: As indicated by the third Reviewer, information related to the flow cytometry algorithmic analysis are also necessary to fulfill this requirement.] 

Additional Editor Comments (if provided):

Given the nature of the results (i.e., negative), small sample size, limited statistical power, and apparent lack of study replication, study limitations need to be described in the Discussion section of the manuscript.

**Minor concerns** -

Abstract line 22 - Remove the word "the" from in between "alters" and "immune"Methods, general - Recognizing that systematic review is becoming common practice and that systematic reviews involve an assessment of study quality, I recommend that the authors provide more details in the Methods section.  For example, you should specify the animal husbandry conditions, if the animals were randomized (and how), test material purity, test material source (city, state), and whether, or not all experimental groups were "exposed" concurrently or if the treatments were staggered over time. Methods, line 189 - Consider adding what the samples were analyzed individually for (i.e., analyzed individually for cytokines (see below).Results, line 249 - Provide a reference, please.Results, line 257 - Change "significant" to "significance"Results, line 270 - 272 - I think it would help the reader if you defined what constitutes a positive result.Figures, general - The text is blurry and difficult to read for several of the figures.  Please adjust the resolution.Figure, legends - I think it would help if you were clear about which mice you were evaluating in the bold text component of each figure legend.  For example, in Figure 1 you specified offspring, but in Figures 2, 3, 4, and 6 the specific animals you are evaluating isn't clearly stated upfront.  The remaining figures are specific to adults and, again, I think it would be better to make that clear upfront.Figure 3 - For consistency with Figures 1, 2, and S1 and ease of reading, please put the figure legend in the top panel.

Reviewers' comments:

Reviewer's Responses to Questions

**Comments to the Author**

1. Is the manuscript technically sound, and do the data support the conclusions?

Reviewer #1: Yes

Reviewer #2: Yes

Reviewer #3: No

2. Has the statistical analysis been performed appropriately and rigorously? 

Reviewer #1: Yes

Reviewer #2: Yes

Reviewer #3: No

3. Have the authors made all data underlying the findings in their manuscript fully available?

Reviewer #1: Yes

Reviewer #2: Yes

Reviewer #3: No

4. Is the manuscript presented in an intelligible fashion and written in standard English?

Reviewer #1: Yes

Reviewer #2: Yes

Reviewer #3: Yes

5. Review Comments to the Author

Reviewer #1: This manuscript by McCall and colleagues looks at the effect of prenatal cadmium (Cd) exposure on development of autoimmunity in mouse models of type 1 diabetes and arthritis. The takeaway from these studies are that prenatal Cd exposure had little effect on development of specific autoimmune phenotypes in these experimental models.

This is essentially a negative study but also a cautionary tale demonstrating that xenobiotic exposures linked to autoimmunity in both humans and animals (mainly mice) can fail to impact disease development in animal models prone to specific diseases. This is known for other xenobiotics, especially after exposure in adults, and so reporting its occurrence for Cd in prenatal exposure is important.

The authors should provide some comparative discussion on the effects of Cd exposure in adult animals as this is known to elicit autoimmune responses. Interestingly, Cd can induce autoantibodies in healthy mice as well as exacerbate autoimmunity in autoimmune prone strains.

In some situations xenobiotic exposure may not exacerbate or accelerate a specific autoimmune phenotype, but may still elicit a response characteristic of that xenobiotic. As indicated above, Cd can induce and/or exacerbate anti-nuclear antibodies (ANA). Thus one aspect worth considering is whether the NOD and SKG mice had increases in ANA following prenatal Cd exposure. This would help establish whether this Cd induced response in adult mice also occurs following prenatal exposure in different models of autoimmunity.

Minor points

1) The text within the figures lacks adequate resolution.

2) The x axis in Figure 2 needs text to identify the groups.

3) In Figure 3 there is no indication of any differences in TNF-a, but the control/non-diabetic mean is much greater than the other groups. This stands in contrast to the IL-17F values where seemingly smaller differences are statistically different.

4) The changes described in the micro-CT images in Figure 5 should be indicated by arrows or circles.

Reviewer #2: Based on previously results, authors investigated if Cd exposure at environmental level could can favor the onset and severity of autoimmune diseases in two animal models. The effect of prenatal Cd (10 ppm) on the development of diabetes (NOD mice) and arthritis (zymosan-induced autoimmune arthritis in SKG mice) were investigated. Small differences were observed in exposed animals compared to control mice, that were judged not biological relevant, leading to the conclusion that prenatal exposure to Cd did not increase the propensity to develop autoimmune diseases.

Below some concerns:

- In the introduction line 78, the nature of the persistent changes in thymus and spleen should be better defined.

- In some group the number of animal is very low (n=4), how can this affect the results? why only 4 animals? taking into account the biological variability in the response, has a statistical analysis been made to define the optimal experimental design to highlight differences and establish the appropriate number of animals?

- Considering that Cd is an environmental contaminant, why exposure did not continue after weaning? In real life, we can imagine continued exposure.

- Figure 1B. In the legend it is not reported at what time point the percent diabetic is referred to. Likely 18-22 weeks, but this should be specified. Interesting is the gender difference, with females exposed to Cd showing a delay in the onset. In this model, if we want to see some effect of cadmium, at most we can hypothesize a protective effect.

- Figure 2. How can authors explain the lack of decrease in Treg previously observed? How can this be transposed to humans? Are the percentage the same as well?

- Figure 4. In this model, it seems an earlier onset in males exposed to Cd.

- Where the levels of anti-inflammatory cytokines affected?

- In the results sections, the nature of changes must be specified.

- In the Discussion, repetition of the results should be avoided.

- Line 402, what do the authors mean with ‘disrupt’?

Reviewer #3: There is as major inconsistency between the text and the data for cytokine production. In Figure 2, the levels of IL-17A and IL-17F show no obvious changes with Cd treatment, whereas TNFalpha levels in male mice are approximately ten-fold lower in non-diabetic mice after Cd treatment. The significance values are shown for two comparisons for IL-17F, but the much larger TNF difference is ignored. This mismatch between the text and the data needs to be resolved.

Publication of negative data is important to provide other investigators with planning information, and to reduce duplication of results. However, in contrast to positive results, it is difficult to identify a statistical significance for a lack of effect. Thus the number of replicates, and the consistency of the results, become more important when reporting negative results. This study used relatively small groups of mice, and there is no indication of whether the experiments were repeated.

In a previous study (ref. 11) on prenatal exposure of mice to Cd, this group found an increase in antibody response and IFNg production, and a decrease in nTreg cells, using two or three experiments with 5-8 mice per condition each. However, in the current study in the NOD mouse, a significant decrease in nTreg cells was not observed. This could have been due to differences between the mouse strains, but it could also be related to the small numbers of mice in the current study. Specifically, in Figure 5 of ref 11, there is a decrease of nTreg cells from about 1.65% to 1.3% with Cd-treated females (males changed in the opposite direction). In the current study, the mean differences were larger, i.e. in Fig. 2, females showed about 3% and 2.1% nTregs with and without Cd. Thus the averages in the two studies are concordant for females (not males) but the authors draw different conclusions because of the increased spread of the data in the current study. These results are a good example of the way that a lack of significant effect may be due to low sample numbers. Phrased differently, the current study does not appear to have been powered sufficiently to detect changes at the level shown in the previous study.

Given that in their previous study, the authors found an impressive difference in IFNg levels after prenatal cadmium exposure, why was IFNg not measured in this study?

Demonstration of the detailed flow cytometry algorithmic analysis (or manual gating) that was used to enumerate cell populations should be provided as supplementary data. This is important for any flow cytometry data, but particularly for Treg cells that have less well-defined marker staining compared to other T cell markers.

At the end of the introduction, the authors state that “unfortunately” Cd did not increase automimmunity in their model. While it is reasonable that the authors are disappointed in the outcome of their hypothesis, at face value their model may suggest that Cd does not have a strong in utero effect on autoimmune disease, which could be good news?

6. PLOS authors have the option to publish the peer review history of their article (what does this mean?). If published, this will include your full peer review and any attached files.

Reviewer #1: **Yes: **K. Michael Pollard

Reviewer #2: No

Reviewer #3: No

---

## [Author Response · Author response to Decision Letter 0]

3 Aug 2021

File naming and author affiliations have been updated

2. To comply with PLOS ONE submissions requirements, in your Methods section, please provide additional information on the animal research and ensure you have included details on methods of anesthesia and/or analgesia, and efforts to alleviate suffering.

Additional information was included. 

All figures are included in the manuscript or uploaded to Figshare and cited.

4. PLOS ONE requires that the data underlying each figure and table be made available to the public prior to publication 

 All raw data used to generate graphs presented as means are included in new table “S1 Table.” Flow cytometry gating strategies are found in “S1 Fig.”

Additional Editor Comments (if provided):

• Given the nature of the results (i.e., negative), small sample size, limited statistical power, and apparent lack of study replication, study limitations need to be described in the Discussion section of the manuscript.

 Added samples to the bone microarchitecture experiments. Text has been updated to clarify the replicates and study limitations have been added to the discussion. 

Minor concerns -

• Abstract line 22 - Remove the word "the" from in between "alters" and "immune" Fixed

• Methods, general - Recognizing that systematic review is becoming common practice and that systematic reviews involve an assessment of study quality, I recommend that the authors provide more details in the Methods section. For example, you should specify the animal husbandry conditions, if the animals were randomized (and how), test material purity, test material source (city, state), and whether, or not all experimental groups were "exposed" concurrently or if the treatments were staggered over time. Added details

• Methods, line 189 - Consider adding what the samples were analyzed individually for (i.e., analyzed individually for cytokines (see below). Added text to clarify

• Results, line 249 - Provide a reference, please. Added

• Results, line 257 - Change "significant" to "significance" Fixed

• Results, line 270 - 272 - I think it would help the reader if you defined what constitutes a positive result. Additional text was added to clarify

• Figures, general - The text is blurry and difficult to read for several of the figures. Please adjust the resolution. Use PACE to upload figures. 

• Figure, legends - I think it would help if you were clear about which mice you were evaluating in the bold text component of each figure legend. For example, in Figure 1 you specified offspring, but in Figures 2, 3, 4, and 6 the specific animals you are evaluating isn't clearly stated upfront. The remaining figures are specific to adults and, again, I think it would be better to make that clear upfront. Legends were updated to clarify animals analyzed

• Figure 3 - For consistency with Figures 1, 2, and S1 and ease of reading, please put the figure legend in the top panel. Changed

Reviewers' comments:

Reviewer's Responses to Questions

Comments to the Author

1. Is the manuscript technically sound, and do the data support the conclusions?

Reviewer #1: Yes

Reviewer #2: Yes

Reviewer #3: No

2. Has the statistical analysis been performed appropriately and rigorously? 

Reviewer #1: Yes

Reviewer #2: Yes

Reviewer #3: No

3. Have the authors made all data underlying the findings in their manuscript fully available?

Reviewer #1: Yes

Reviewer #2: Yes

Reviewer #3: No

4. Is the manuscript presented in an intelligible fashion and written in standard English?

Reviewer #1: Yes

Reviewer #2: Yes

Reviewer #3: Yes

5. Review Comments to the Author

Reviewer #1: This manuscript by McCall and colleagues looks at the effect of prenatal cadmium (Cd) exposure on development of autoimmunity in mouse models of type 1 diabetes and arthritis. The takeaway from these studies are that prenatal Cd exposure had little effect on development of specific autoimmune phenotypes in these experimental models.

This is essentially a negative study but also a cautionary tale demonstrating that xenobiotic exposures linked to autoimmunity in both humans and animals (mainly mice) can fail to impact disease development in animal models prone to specific diseases. This is known for other xenobiotics, especially after exposure in adults, and so reporting its occurrence for Cd in prenatal exposure is important.

The authors should provide some comparative discussion on the effects of Cd exposure in adult animals as this is known to elicit autoimmune responses. Interestingly, Cd can induce autoantibodies in healthy mice as well as exacerbate autoimmunity in autoimmune prone strains.

In some situations xenobiotic exposure may not exacerbate or accelerate a specific autoimmune phenotype, but may still elicit a response characteristic of that xenobiotic. As indicated above, Cd can induce and/or exacerbate anti-nuclear antibodies (ANA). Thus one aspect worth considering is whether the NOD and SKG mice had increases in ANA following prenatal Cd exposure. This would help establish whether this Cd induced response in adult mice also occurs following prenatal exposure in different models of autoimmunity. Performed ANA screen in SKG sera.

Minor points

1) The text within the figures lacks adequate resolution. Upload using PACE

2) The x axis in Figure 2 needs text to identify the groups. Removed ticks on the x-axis so they didn’t distract from the legend. 

3) In Figure 3 there is no indication of any differences in TNF-a, but the control/non-diabetic mean is much greater than the other groups. This stands in contrast to the IL-17F values where seemingly smaller differences are statistically different. The data originally in the graph were incorrect. The stats were performed on the raw data. The graph has been updated with the correct data. 

4) The changes described in the micro-CT images in Figure 5 should be indicated by arrows or circles. Added

Reviewer #2: Based on previously results, authors investigated if Cd exposure at environmental level could can favor the onset and severity of autoimmune diseases in two animal models. The effect of prenatal Cd (10 ppm) on the development of diabetes (NOD mice) and arthritis (zymosan-induced autoimmune arthritis in SKG mice) were investigated. Small differences were observed in exposed animals compared to control mice, that were judged not biological relevant, leading to the conclusion that prenatal exposure to Cd did not increase the propensity to develop autoimmune diseases.

Below some concerns:

- In the introduction line 78, the nature of the persistent changes in thymus and spleen should be better defined. Clarified reduction in nTregs in text.

- In some group the number of animal is very low (n=4), how can this affect the results? why only 4 animals? taking into account the biological variability in the response, has a statistical analysis been made to define the optimal experimental design to highlight differences and establish the appropriate number of animals? Two mice from each group were added to the analysis. microCT data was performed on a representative subset of animals that were scored using the arthritic index method. Both methods showed no difference in disease severity. 

- Considering that Cd is an environmental contaminant, why exposure did not continue after weaning? In real life, we can imagine continued exposure. This is something that can be addressed in the future; however, for this study we wanted to specifically test the in utero exposure. 

- Figure 1B. In the legend it is not reported at what time point the percent diabetic is referred to. Likely 18-22 weeks, but this should be specified. Interesting is the gender difference, with females exposed to Cd showing a delay in the onset. In this model, if we want to see some effect of cadmium, at most we can hypothesize a protective effect. Added discussion regarding “protection” in the discussion. 

- Figure 2. How can authors explain the lack of decrease in Treg previously observed? How can this be transposed to humans? Are the percentage the same as well? Clarified that these are different mouse models.

- Figure 4. In this model, it seems an earlier onset in males exposed to Cd. This is consistent with the published data of the model. 

- Where the levels of anti-inflammatory cytokines affected? Unfortunately, these were not measured.

- In the results sections, the nature of changes must be specified. Changed ambiguous “differences” to reduced or increased.

- In the Discussion, repetition of the results should be avoided. Moved specific data and p values to result.

- Line 402, what do the authors mean with ‘disrupt’? clarified

Reviewer #3: There is as major inconsistency between the text and the data for cytokine production. In Figure 2, the levels of IL-17A and IL-17F show no obvious changes with Cd treatment, whereas TNFalpha levels in male mice are approximately ten-fold lower in non-diabetic mice after Cd treatment. The significance values are shown for two comparisons for IL-17F, but the much larger TNF difference is ignored. This mismatch between the text and the data needs to be resolved. The data originally in the graph were incorrect. The stats were performed on the raw data. The graph has been updated with the correct data.

Publication of negative data is important to provide other investigators with planning information, and to reduce duplication of results. However, in contrast to positive results, it is difficult to identify a statistical significance for a lack of effect. Thus the number of replicates, and the consistency of the results, become more important when reporting negative results. This study used relatively small groups of mice, and there is no indication of whether the experiments were repeated. Clarified repeats in methods.

In a previous study (ref. 11) on prenatal exposure of mice to Cd, this group found an increase in antibody response and IFNg production, and a decrease in nTreg cells, using two or three experiments with 5-8 mice per condition each. However, in the current study in the NOD mouse, a significant decrease in nTreg cells was not observed. This could have been due to differences between the mouse strains, but it could also be related to the small numbers of mice in the current study. Specifically, in Figure 5 of ref 11, there is a decrease of nTreg cells from about 1.65% to 1.3% with Cd-treated females (males changed in the opposite direction). In the current study, the mean differences were larger, i.e. in Fig. 2, females showed about 3% and 2.1% nTregs with and without Cd. Thus the averages in the two studies are concordant for females (not males) but the authors draw different conclusions because of the increased spread of the data in the current study. These results are a good example of the way that a lack of significant effect may be due to low sample numbers. Phrased differently, the current study does not appear to have been powered sufficiently to detect changes at the level shown in the previous study.

Included figure without dividing the mice by disease state to show that Cd itself is not inducing Treg changes in the model. Increased animal numbers where possible and added additional assays. 

Given that in their previous study, the authors found an impressive difference in IFNg levels after prenatal cadmium exposure, why was IFNg not measured in this study? Added paragraph in discussion

Demonstration of the detailed flow cytometry algorithmic analysis (or manual gating) that was used to enumerate cell populations should be provided as supplementary data. This is important for any flow cytometry data, but particularly for Treg cells that have less well-defined marker staining compared to other T cell markers. Added gating strategy in supplemental figure 1. 

At the end of the introduction, the authors state that “unfortunately” Cd did not increase automimmunity in their model. While it is reasonable that the authors are disappointed in the outcome of their hypothesis, at face value their model may suggest that Cd does not have a strong in utero effect on autoimmune disease, which could be good news? Updated introduction to include the positive aspect of the data. 

---

## [Decision Letter · Decision Letter 1]

24 Aug 2021

Prenatal cadmium exposure does not induce greater incidence or earlier onset of autoimmunity in the offspring

PONE-D-21-08667R1

Dear Dr. Barnett,

We’re pleased to inform you that your manuscript has been judged scientifically suitable for publication and will be formally accepted for publication once it meets all outstanding technical requirements.

Kind regards,

David M. Lehmann, Ph.D.

Academic Editor

PLOS ONE

Additional Editor Comments (optional):

Reviewers' comments:

Reviewer's Responses to Questions

**Comments to the Author**

1. If the authors have adequately addressed your comments raised in a previous round of review and you feel that this manuscript is now acceptable for publication, you may indicate that here to bypass the “Comments to the Author” section, enter your conflict of interest statement in the “Confidential to Editor” section, and submit your "Accept" recommendation.

Reviewer #1: All comments have been addressed

Reviewer #2: All comments have been addressed

2. Is the manuscript technically sound, and do the data support the conclusions?

Reviewer #1: Yes

Reviewer #2: Yes

3. Has the statistical analysis been performed appropriately and rigorously? 

Reviewer #1: Yes

Reviewer #2: Yes

4. Have the authors made all data underlying the findings in their manuscript fully available?

Reviewer #1: Yes

Reviewer #2: Yes

5. Is the manuscript presented in an intelligible fashion and written in standard English?

Reviewer #1: Yes

Reviewer #2: Yes

6. Review Comments to the Author

Reviewer #1: (No Response)

Reviewer #2: I thank the Authors. I have no further comments, as Authors have properly addressed all my concerns.

7. PLOS authors have the option to publish the peer review history of their article (what does this mean?). If published, this will include your full peer review and any attached files.

Reviewer #1: **Yes: **Kenneth Michael Pollard

Reviewer #2: No

---

## [Editor Report · Acceptance letter]

26 Aug 2021

PONE-D-21-08667R1 

Prenatal cadmium exposure does not induce greater incidence or earlier onset of autoimmunity in the offspring 

Dear Dr. Barnett:

I'm pleased to inform you that your manuscript has been deemed suitable for publication in PLOS ONE. Congratulations! Your manuscript is now with our production department. 

Kind regards, 

on behalf of

Dr. David M. Lehmann 

Academic Editor

PLOS ONE